# Costs of community-wide mass drug administration and school-based deworming for soil-transmitted helminths: evidence from a randomised controlled trial in Benin, India and Malawi

Chloe Morozoff [1], Euripide Avokpaho,[2]
Saravanakumar Puthupalayam Kaliappan,[3] James Simwanza [4]
Samuel Paul Gideon,[3] Wongani Lungu,[4] Parfait Houngbegnon,[2]
Katya Galactionova [5,6] Maitreyi Sahu,[1] Khumbo Kalua,[4] Adrian J F Luty,[7]
Moudachirou Ibikounlé,[2,8] Robin Bailey,[9] Rachel Pullan,[10]
Sitara Swarna Rao Ajjampur,[3] Judd Walson,[1] Arianna Rubin Means [1]

Correspondence to
Dr Arianna Rubin Means;
aerubin@uw.edu

## ABSTRACT

**Objectives** Current guidelines for the control of soil-transmitted helminths (STH) recommend deworming children and other high-risk groups, primarily using school-based deworming (SBD) programmes. However, targeting individuals of all ages through community-wide mass drug administration (cMDA) may interrupt STH transmission in some settings. We compared the costs of cMDA to SBD to inform decision-making about future updates to STH policy.

**Design** We conducted activity-based microcosting of cMDA and SBD for 2 years in Benin, India and Malawi within an ongoing cMDA trial.

**Setting** Field sites and collaborating research institutions.

**Primary and secondary outcomes** We calculated total financial and opportunity costs and costs per treatment administered (unit costs in 2019 USD ($)) from the service provider perspective, including costs related to community drug distributors and other volunteers.

**Results** On average, cMDA unit costs were more expensive than SBD in India ($1.17 vs $0.72) and Malawi ($2.26 vs $1.69), and comparable in Benin ($2.45 vs $2.47). cMDA was more expensive than SBD in part because most costs (~60%) were 'supportive costs' needed to deliver treatment with high coverage, such as additional supervision and electronic data capture. A smaller fraction of cMDA costs (~30%) was routine expenditures (eg, drug distributor allowances). The remaining cMDA costs (~10%) were opportunity costs of staff and volunteer time. A larger percentage of SBD costs was opportunity costs for teachers and other government staff (between ~25% and 75%). Unit costs varied over time and were sensitive to the number of treatments administered.

**Conclusions** cMDA was generally more expensive than SBD. Accounting for local staff time (volunteers, teachers, health workers) in community programmes is important and drives higher cost estimates than commonly recognised in the literature. Costs may be lower outside of

## STRENGTHS AND LIMITATIONS OF THIS STUDY

⇒ We used rigorous microcosting methods to collect costs associated with community-wide mass drug administration and school-based deworming and corresponding treatment data, in real time.

⇒ The granularity of data collected provides rich information regarding the resource needs for deworming programmes, and how these may vary across countries and delivery modalities (school vs community-based treatment).

⇒ We estimated opportunity costs of the volunteer workforce and currently employed government staff (eg, teachers, community drug distributors, supervisors), which are often excluded from deworming costing studies.

⇒ Although costs associated with research and trial administration were not included in this study, it is possible that some costs (eg, programme management, planning and supervision) may be higher in this research setting than what would be observed in routine deworming programmes.

a trial setting, given a reduction in supportive costs used to drive higher treatment coverage and economies of scale.

**Trial registration number** NCT03014167.

## INTRODUCTION

Soil-transmitted helminths (STH) are a group of intestinal parasites (*Ascaris lumbricoides*, *Ancylostoma duodenale*, *Necator americanus* and *Trichuris trichiura*) that globally affect approximately 1.5 billion individuals annually, predominantly in sub-Saharan Africa, East Asia and Latin America.[1] Moderate-to-heavy infection with STH is associated with diarrhoea, malnutrition, anaemia, wasting,



stunting and cognitive delay.[1 2] To reduce the burden of STH morbidity, the WHO targets elimination of STH as a public health problem by 2030.[3] Current STH control guidelines recommend preventative chemotherapy (deworming using albendazole or mebendazole) for high-risk populations such as children, non-pregnant adolescent girls and women of reproductive age.[2]

STH control programmes include annual or biannual school-based deworming (SBD), where teachers and health workers deliver preventative chemotherapy to preschool and school-aged children.[2] SBD is a low-cost intervention; SBD leverages existing infrastructure (schools) as a delivery platform while drug costs are low due to global drug donation programmes.[4] A review of STH treatment costs estimates SBD costs at $0.30 (2015 USD) per child treated, much lower than the cost of screening and treating a single individual for STH annually ($4.89/person in 2015 USD).[2] Costs of deworming preschool-aged children or other community members outside of schools is estimated at $0.63 (2015 USD) per person treated.[2] Although SBD is a low-cost intervention for controlling STH, non-school attending children may be missed by these programmes and reinfection of children within the community from adult reservoirs may require continuous treatment.[5]

It may be possible to interrupt STH transmission by expanding deworming eligibility to individuals of all ages.[6 7] The DeWorm3 project is an ongoing cluster-randomised trial testing the feasibility of interrupting STH transmission using community-wide mass drug administration (cMDA) in Benin, India and Malawi.[8] If successful, scaling-up cMDA would require evidence on the relative cost compared with standard-of-care SBD. Although studies have evaluated the costs of mass drug administration for neglected tropical diseases, costs vary based on country implementation strategy (eg, use of volunteers or salaried staff), disease control programme,

age of programme, size of population treated and costing methods used.[9–16] To our knowledge, there are no studies that directly compare costs of cMDA and SBD for STH control, within the same setting and methodological framework.

This study systematically identified, measured and compared resources for implementation of 12 rounds of cMDA and 8 rounds of SBD across the DeWorm3 sites, during implementation of the trial. Determining the costs and cost drivers of expanding STH treatment to all individuals in a community will be essential for shaping future STH policy.

## METHODS
### Overview of DeWorm3

The DeWorm3 project was implemented in Come Commune of Benin, Tamil Nadu State of India and Mangochi District of Malawi. These sites were selected because they had previously implemented lymphatic filariasis programmes over five or more rounds with albendazole coadministered with ivermectin or diethylcarbamazine.[8] In each site, 20 control clusters (minimum population size of 1650 persons per cluster) were randomised to SBD (either annually or biannually, per the country's standard of care) and 20 intervention clusters were randomised to biannual cMDA. In intervention clusters, SBD continued to be implemented as per the country's standard of care but was not costed; during treatment rounds in which SBD was also implemented, cMDA was conducted after SBD (see figure 1). During the DeWorm3 project, cMDA and SBD were implemented for 3 years, from 2018 to 2020. During SBD, teachers distributed albendazole to children, with support from community health workers, known as community drug distributors (CDDs) in Benin, Accredited Social Health Activists (ASHAs) in India and Health Surveillance

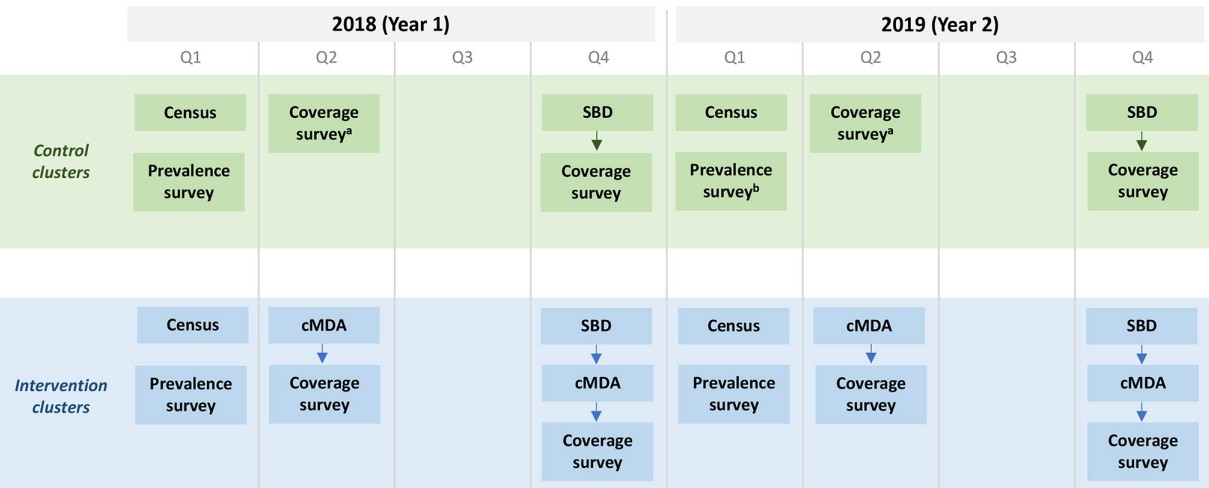

**Figure 1** Flow of DeWorm3 activities conducted in intervention and control clusters. Activities include: census, prevalence survey, school-based deworming, community-wide mass drug administration, and coverage survey. Acronyms: quarter (Q), school-based deworming (SBD), community-wide mass drug administration (cMDA). [a] In India, SBD is also conducted in quarter 2, prior to the coverage survey. [b] In Malawi, no prevalence survey was conducted in year 2.

Assistants (HSAs) in Malawi. During cMDA, albendazole was delivered door-to-door in the community by community health workers (CDDs in Benin, CDDs and ASHAs in India, HSAs in Malawi) alongside electronic data collectors, referred to as enumerators. For both treatment strategies, supervision was conducted by DeWorm3 and government staff. Number of schools, villages and other site-level contextual attributes are described in online supplemental appendix 1 and 2.

In addition to deworming, DeWorm3 also conducted an annual census to enumerate the full population in study catchment areas, annual prevalence surveys to determine STH prevalence and intensity and post-MDA coverage surveys to assess the reach of cMDA and SBD (figure 1).[8] These activities are not consistently conducted in national deworming programmes but could be indicated in future elimination programmes that require more intensive monitoring and evaluation.

## Costing study design

We conducted activity-based microcosting from the service provider perspective (Ministry of Health and/or Education) during the first 2 years of DeWorm3 implementation in order to explore heterogeneity in costs across rounds of implementation. Across the three sites, we conducted 2 years of intensive microcosting, resulting in data from 12 rounds of cMDA and 8 rounds of SBD. Costing the first year of cMDA implementation allowed us to capture costs related to start-up, while the second year provided a more accurate portrayal of costs related to routine implementation. This analysis includes opportunity costs associated with all health worker involvement in implementation, including teachers and community volunteers engaging in drug delivery. Costs to the household were not assessed as they are assumed to be negligible.[10 11] We measured all resources required to deliver cMDA and SBD in DeWorm3 clusters, resulting in over 8000 data points, and converted their value into a cost estimate (including borrowed and donated resources).[17 18]

The methodology for costing cMDA is detailed in Galactionova *et al*,[19] and additional SBD data collection tools are detailed in online supplemental appendix 2. We briefly describe the strategy used. DeWorm3 staff in each site recorded resource use and costs related to the implementation of trial activities within an Excel-based costing tool. Data were collected in real-time and were entered separately for activities including programme management (overheads), planning and each round of the census, prevalence survey, SBD, cMDA and coverage survey. Within the tool, we also quantified borrowed resources used, such as borrowed vehicles and volunteer time. Other data sources were used to collect or allocate costs not included in the costing tool, such as government expenditures (see online supplemental appendix 2).

Following data collection, all costing data were iteratively reviewed for quality and completeness. Costs related to DeWorm3 research only (eg, qualitative research or school surveys) were not included in the data collection

instruments and, if identified, were removed during data cleaning.[20]

## Analysing financial and opportunity costs

Financial costs included actual expenditures on goods and services purchased by the DeWorm3 project or site governments. We analysed these data in Stata (V.16.1). Costs were converted to USD using the annual average exchange rate based on the year in which the costs were incurred.[21] When costs were shared across multiple activities—such as vehicles or personnel salaries—we allocated costs based on the number of days required to implement each subactivity. We allocated costs reported at the district or state level via government budgets to the DeWorm3 study area using population proportionate to size estimates. We annualised startup costs over the 3-year duration of cMDA and SBD implementation, and capital items based on their useful life years, using a 3% discount rate.[18 22] All costs are presented in 2019 USD; costs incurred before 2019 were inflated using gross domestic product implicit price deflators.[23 24] Costs in local currency are presented in online supplemental appendix 3.

Opportunity costs included the costs of donated drugs, volunteer time (CDDs, ASHAs and community volunteers) and time costs for currently employed government staff. We estimated costs associated with volunteer time spent delivering drugs using the DeWorm3 trial's digital treatment forms (described in online supplemental appendix 2). We used country-specific average earnings to estimate the opportunity costs associated with volunteer time (2010–2011 regional annual salary adjusted to relevant year using annual growth rate in India and 2018 national monthly earnings in Benin and Malawi).[25 26] For government staff (eg, national and district-level personnel, teachers and health centre staff), we collected salaries through Ministry of Health costing surveys. We derived government staff time spent on activities from costing data collection tools, and teacher time spent on SBD from a school survey. We calculated total economic costs (financial plus opportunity costs) per site, per year and by activity, subactivity and input classification. Key costing inputs such as the number of implementing staff, average salaries and allowances are described in online supplemental appendix 2.

## Estimating routine and supportive programme costs

Because the DeWorm3 Project included several activities related to the delivery and monitoring of MDA that may not be present in all deworming programmes, we classified and distinguished costs as either routine MDA programme costs or supportive programme costs. Routine programme costs included activities typically implemented by a government (eg, training of CDDs). Supportive costs included additional activities aimed at optimising coverage and compliance. For example, electronic data were collected to monitor cMDA coverage in real-time and identify areas in need of additional

sensitisation and mop-up. In general, supportive activities included: (1) start-up planning costs, (2) additional supervision from a non-governmental organisation (NGO) implementing partner, (3) additional sensitisation activities, (4) electronic data collection and (5) programme management costs associated with these supportive activities. Additional details regarding routine and supportive costs are presented in online supplemental appendix 2.

## Unit cost analysis

The cost per treatment administered (ie, unit cost) was determined by dividing costs per round by the total number of treatments administered. The number of treatments administered via cMDA was abstracted from MDA treatment forms (household-level forms completed by enumerators during cMDA). The number of treatments administered via SBD was estimated from paper SBD forms filled out by school and/or DeWorm3 field staff, then transferred to an electronic format. One and two-way sensitivity analyses were conducted to explore how the average cost per treatment administered would change due to variation in key costing inputs and coverage levels (methods described in online supplemental appendix 4).

## Patient and public involvement

Community members living in STH endemic areas were not involved in the design, conduct, reporting or dissemination of this costing study. Ministry of Health and Education staff were involved in the conduct of this costing study (including data collection and dissemination) and in the design and conduct of the wider DeWorm3 trial.

## RESULTS

### Total costs of cMDA and SBD

Between February 2018 and December 2019, a total of 12 rounds of cMDA and 8 rounds of SBD were delivered across DeWorm3 sites in Benin, India and Malawi. Table 1 details the number of treatments administered, total costs and unit costs across treatment strategies, sites and rounds. The total number of treatments administered for a given round of MDA ranged from 9298 (Benin SBD round 2) to 57398 (India cMDA round 4). Total costs of SBD ranged from $12763 in India (round 4) to $25933 in Benin (round 4), while total costs of cMDA ranged from $61806 (India, round 4) to $129369 (Malawi, round 1). cMDA unit costs varied from $1.08 in India (round 4) to $2.90 in Benin (round 4). Within sites, cMDA unit costs varied across the four rounds, fluctuating by $0.73 in Benin and Malawi and $0.21 in India. SBD was generally less expensive than cMDA, with approximately one-third the number of treatments administered and one-quarter

**Table 1** Total economic costs and number of treatments administered through community-wide mass drug administration and school-based deworming, per country-round, in 2019 USD ($)

| Metric | Benin | | India | | Malawi | |
|---|---|---|---|---|---|---|
| | **cMDA** | **SBD** | **cMDA** | **SBD** | **cMDA** | **SBD** |
| Number of treatments administered* | | | | | | |
| Round 1 | 45280 | – | 55953 | 15266 | 49518 | – |
| Round 2 | 37913 | 9298 | 55758 | 19152 | 38641 | 16077 |
| Round 3 | 42398 | – | 57353 | 21396 | 52122 | – |
| Round 4 | 32529 | 10343 | 57398 | 20586 | 49709 | 12964 |
| Total costs† | | | | | | |
| Round 1 | 106695 | – | 71969 | 13854 | 129369 | – |
| Round 2 | 82287 | 22516 | 64416 | 14089 | 97512 | 23251 |
| Round 3 | 99664 | – | 66129 | 12794 | 97838 | – |
| Round 4 | 94422 | 25933 | 61806 | 12763 | 100112 | 24812 |
| Cost per treatment administered | | | | | | |
| Round 1 | 2.36 | – | 1.29 | 0.91 | 2.61 | – |
| Round 2 | 2.17 | 2.42 | 1.16 | 0.74 | 2.52 | 1.45 |
| Round 3 | 2.35 | – | 1.15 | 0.60 | 1.88 | – |
| Round 4 | 2.90 | 2.51 | 1.08 | 0.62 | 2.01 | 1.91 |

Note: Dashes (–) represent situations where no data were collected. SBD was only implemented annually in Benin and Malawi, so no data were available for rounds 1 and 3.
*Treatments administered for cMDA include all eligible individuals who received treatment by DeWorm3 through cMDA in the intervention clusters (source: DeWorm3 MDA treatment logs). Population treated for SBD includes all children treated in schools within the DeWorm3 control clusters (source: SBD treatment logs).
†Total costs include both financial and opportunity costs.
cMDA, community-wide mass drug administration; SBD, school-based deworming.

**Table 2** Average unit costs (2019 USD ($)) for community-wide mass drug administration across 2 years

| | Benin* | India* | Malawi* |
|---|---|---|---|
| **Planning** | **$ 0.10** | **$ 0.04** | **$ 0.01** |
| Supportive (financial) | $ 0.10 | $ 0.04 | $ 0.01 |
| **Programme management** | **$ 0.63** | **$ 0.40** | **$ 0.50** |
| Routine (financial) | $ 0.28 | $ 0.16 | $ 0.15 |
| Routine (opportunity)—time costs for government staff† | $ 0.01 | – | < $ 0.01 |
| Supportive (financial) | $ 0.34 | $ 0.24 | $ 0.35 |
| **Community sensitisation** | **$ 0.24** | **$ 0.17** | **$ 0.17** |
| Routine (financial) | $ 0.11 | $ 0.02 | $ 0.06 |
| Routine (opportunity)—time costs for government staff and volunteers | $ 0.01 | < $ 0.01 | $ 0.04 |
| Supportive (financial)—additional sensitisation activities | $ 0.01 | < $ 0.01 | $ 0.01 |
| Supportive (financial)—NGO supervision | $ 0.11 | $ 0.14 | $ 0.06 |
| **Training** | **$ 0.34** | **$ 0.11** | **$ 0.26** |
| Routine (financial) | $ 0.12 | $ 0.01 | $ 0.07 |
| Routine (opportunity)—time costs for government staff and volunteers | $ 0.02 | $ 0.03 | $ 0.02 |
| Supportive (financial)—training of electronic data collectors | $ 0.11 | $ 0.05 | $ 0.05 |
| Supportive (financial)—NGO supervision and training support | $ 0.08 | $ 0.02 | $ 0.11 |
| **Drug delivery** | **$ 1.13** | **$ 0.46** | **$ 1.32** |
| Routine (financial) | $ 0.36 | $ 0.07 | $ 0.20 |
| Routine (opportunity)—time costs for government staff and volunteers | $ 0.15 | $ 0.11 | $ 0.18 |
| Routine (opportunity)—donated drugs | $ 0.05 | $ 0.01 | $ 0.05 |
| Supportive (financial)—electronic data capture | $ 0.29 | $ 0.19 | $ 0.31 |
| Supportive (financial)—NGO supervision | $ 0.27 | $ 0.07 | $ 0.58 |
| *Average unit costs‡* | *$ 2.45* | *$ 1.17* | *$ 2.26* |
| *Routine (financial)* | *$ 0.87* | *$ 0.26* | *$ 0.48* |
| *Routine (opportunity)* | *$ 0.26* | *$ 0.16* | *$ 0.30* |
| *Supportive (financial)* | *$ 1.32* | *$ 0.75* | *$ 1.48* |

Note: Dashes (–) represent situations where no costs were observed. Total economic costs are presented, as well as a breakdown of costs by routine versus. supportive activities, and financial vs. opportunity costs.
The bolded costs represent the sum of the indented routine and supportive costs below them. The italicized values summarize total costs from the table.
*Analysis includes 2 years of cMDA. As cMDA was conducted bi-annually in each country, results are presented as the average across four rounds.
†Government staff include supervisory and implementing staff whose salaries are paid by the ministry of health. Examples include: nurses and health officers, HSAs (Malawi only), as well as national and subnational government officials involved in the programme.
‡Routine and supportive activities and related resources are described in online supplemental appendix 2. Financial costs represent actual expenditure on goods and services purchased by the government or NGO implementing partner. Opportunity costs, on the other hand, include costs forgone by using a resource in a particular way. These opportunity costs recognise and value the cost of using resources, as these resources are then unavailable for productive use elsewhere. Opportunity costs in this analysis include: costs of donated albendazole, volunteer time spent on the project (such as volunteer drug distributors) and estimated government staff salary costs.
cMDA, community-wide mass drug administration ; HSA, Health Surveillance Assistant; NGO, non-governmental organisation.

of the total costs. SBD unit costs varied from $0.60 in India (round 3) to $2.51 in Benin (round 4). Within sites, SBD costs fluctuated $0.09 across two rounds in Benin, $0.31 across four rounds in India and $0.46 across two rounds in Malawi. Subactivity costs also varied across rounds, as detailed in online supplemental appendix 5.

## Average unit costs of cMDA and SBD

Activity-specific unit costs for cMDA and SBD are presented in tables 2 and 3, respectively. Average cMDA unit costs were $2.45 in Benin, $1.17 in India and $2.26 in Malawi. Routine financial costs were approximately 20%–35% of unit costs, at $0.87 in Benin, $0.26 in India and $0.48 in Malawi. The majority of routine financial costs (approximately 70%–80%) were allowances for key implementing staff (eg, lunch, travel and/or mobile allowances for CDDs, health centre staff, district and national government supervisors, sensitisation staff). Routine opportunity costs, including donated drugs and

**Table 3** Average unit costs (2019 USD ($)) for school-based deworming across 2 years

| | Benin* | India† | Malawi* |
|---|---|---|---|
| **Planning** | **$ 0.07** | **$ 0.00** | **$ 0.01** |
| Supportive (financial) | $ 0.07 | — | $ 0.01 |
| **Programme management** | **$ 0.69** | **$ 0.19** | **$ 0.40** |
| Routine (financial) | — | — | $ 0.15 |
| Routine (opportunity)—time costs for government staff‡ | $ 0.25 | $ 0.11 | $ 0.00 |
| Supportive (financial) | $ 0.44 | $ 0.08 | $ 0.25 |
| **Community sensitisation** | **$ 0.26** | **$ 0.01** | **$ 0.11** |
| Routine (financial) | $ 0.14 | $ 0.01 | $ 0.04 |
| Routine (opportunity)—time costs for government staff and volunteers | — | — | $ 0.05 |
| Supportive (financial)—additional sensitisation activities | $ 0.05 | — | $ 0.01 |
| Supportive (financial)—NGO supervision | $ 0.07 | — | $ 0.02 |
| **Training** | **$ 0.61** | **$ 0.18** | **$ 0.25** |
| Routine (financial) | $ 0.27 | $ 0.02 | $ 0.08 |
| Routine (opportunity)—time costs for government staff and volunteers | $ 0.20 | $ 0.14 | $ 0.11 |
| Supportive (financial)—training of electronic data collectors | $ 0.06 | $ 0.02 | $ 0.02 |
| Supportive (financial)—NGO supervision and training support | $ 0.08 | $ 0.01 | $ 0.04 |
| **Drug delivery** | **$ 0.83** | **$ 0.33** | **$ 0.91** |
| Routine (financial) | $ 0.12 | $ 0.01 | $ 0.22 |
| Routine (opportunity)—time costs for government staff and volunteers | $ 0.56 | $ 0.28 | $ 0.17 |
| Routine (opportunity)—donated drugs | $ 0.06 | $ 0.01 | $ 0.06 |
| Supportive (financial)—electronic data capture | $ 0.02 | $ 0.02 | $ 0.21 |
| Supportive (financial)—NGO supervision | $ 0.07 | $ 0.01 | $ 0.25 |
| ***Average unit costs§*** | ***$ 2.47*** | ***$ 0.72*** | ***$ 1.69*** |
| *Routine (financial)* | *$ 0.53* | *$ 0.03* | *$ 0.48* |
| *Routine (opportunity)* | *$ 1.07* | *$ 0.54* | *$ 0.40* |
| *Supportive (financial)* | *$ 0.87* | *$ 0.14* | *$ 0.81* |

Note: Dashes (–) represent situations where no costs were observed. Total economic costs are presented, as well as a breakdown of costs by routine program vs. supportive program activities, and financial vs. opportunity costs.
The bolded costs represent the sum of the indented routine and supportive costs below them. The italicized values summarize total costs from the table.
*Analysis includes 2 years of SBD. In India, SBD was conducted bi-annually, so results are presented as the average across four rounds.
†Analysis includes 2 years of SBD. In Malawi and Benin, SBD was conducted annually, so results are presented as the average of two rounds.
‡Government staff include supervisory and implementing staff whose salaries are paid by the Ministry of Health. Examples include: nurses and health officers, teachers, and national and subnational government officials involved in the programme.
§Routine and supportive activities and related resources are described in online supplemental appendix 2. Financial costs represent actual expenditure on goods and services purchased by the government or NGO implementing partner. Opportunity costs, on the other hand, include costs forgone by using a resource in a particular way. These opportunity costs recognise and value the cost of using resources, as these resources are then unavailable for productive use elsewhere. Opportunity costs in this analysis include: costs of donated albendazole, volunteer time spent on the project (such as volunteer drug distributors), and estimated government staff salary costs.
NGO, non-governmental organisation; SBD, school-based deworming.

government and volunteer time, were approximately 10% of unit costs (ranging from $0.16 in India to $0.30 in Malawi). Other routine costs included materials and supplies, equipment or building rentals and vehicle costs for supervision (online supplemental appendix 5). Supportive programme costs, including costs of electronic data collection with additional supervision in the DeWorm3 project, comprised the majority of unit costs (approximately 60%).

On average, SBD unit costs were $2.47 in Benin, $0.72 in India and $1.69 in Malawi. Routine financial costs, such as per-diems, fuel, and materials, were approximately 5%–30% of unit costs and were more expensive in Benin and Malawi ($0.53 and $0.48 respectively) as compared with India ($0.03). Routine opportunity costs, mainly teacher and school-level staff time, represented the largest share of costs in Benin and India (approximately 45% and 75%, respectively); the governments of

Benin and India were the primary SBD implementers. In Malawi, where SBD was delivered by the DeWorm3 team, routine opportunity costs were only one-fourth of costs, whereas supportive activities represented half of unit costs.

Across sites, average unit costs were generally higher for cMDA as compared with SBD, except for Benin. However, routine cMDA costs were consistently less expensive compared with SBD, driven in part by the high opportunity costs of SBD. Across cMDA and SBD, drug delivery followed by programme management were the most expensive activities. Drug delivery included initial drug distribution as well as mop-up activities (approximately 10%–20% of drug delivery costs). The largest resource input was staff wages and per-diems, representing 56%–91% of average unit costs, generally followed by vehicle costs (online supplemental appendix 5). Routine vehicle costs were used for government supervision and transport for training. However, the majority of vehicle costs were used for supportive activities, mainly field staff supervision and transport of enumerators to field sites each day for mobile data collection. Vehicle costs contributed to a higher share of costs in Malawi, compared with other sites. Approximately, 15% of SBD and 25% cMDA costs were fixed or capital costs (online supplemental appendix 5), meaning that the expenses do not depend on the quantity of treatments delivered. Examples of fixed costs include programme overheads such as rent, central staff salaries, etc. When examining how unit costs per subactivity varied across rounds, actual MDA delivery costs were the most variable across sites and rounds, followed by programme management costs (online supplemental appendix 5). After planning costs, which were annualised across rounds, community sensitisation showed the least amount of variability in unit costs across countries, rounds and treatment strategies.

### Additional programmatic costs

Costs of additional activities, such as an annual census, prevalence surveys and coverage surveys are not included within cMDA and SBD unit cost estimates but are detailed in online supplemental appendix 5. In brief, costs of annual censuses ranged from $0.54 (India year 2) to $1.81 (Benin year 1) per person censused. Annual prevalence surveys where stool samples were analysed using Kato-Katz ranged from $11.98 (India year 1) to $28.78 per person surveyed (India year 2); variability in costs was due to cross-country differences and shared laboratory costs in year 1 of the survey. Finally, coverage surveys conducted post-MDA were estimated between $1.33 (India year 1) and $4.64 (Benin year 1) per person surveyed.

### Sensitivity analyses

In one-way and two-way sensitivity analyses (figure 2), the largest changes in cMDA and SBD unit costs were driven by altering coverage rates and supportive costs. Changing coverage rates in Malawi resulted in the largest change in estimated unit costs. Estimated deworming programme coverage rates varied widely across clusters in Malawi (from 19% to 74% for SBD and 64% to 96% for cMDA), resulting in unit costs ranging from $1.26 to $4.91 per SBD treatment administered and $1.93 to $2.87 per cMDA treatment administered. Costs decreased in two-way analyses when supportie costs were removed and coverage rates were reduced to approximately 60% cMDA and SBD coverage (assuming that a reduction in support would result in a reduction in coverage); unit costs decreased by 30% or more in most cases. In these two-way sensitivity analyses, the cost of cMDA and SBD was similar, with a net difference of $0.03 to $0.17. Unit costs did not fluctuate substantially in one-way sensitivity analyses exploring opportunity costs of drugs and volunteer time and two-way sensitivity analyses exploring coverage and sensitisation costs.

### DISCUSSION

Costs and resource needs are important pieces of evidence for governments considering updating standards of care and related policies, such as a potential shift from SBD to cMDA. The DeWorm3 project provided a unique platform to assess and compare the costs of two deworming treatment strategies (SBD and cMDA) across heterogeneous STH-endemic settings. We found the average unit cost per treatment administered to be higher for cMDA compared with SBD in India and Malawi, and comparable in Benin.

Costs of MDA for neglected tropical diseases (NTDs), including deworming, vary considerably in the literature, depending on treatment strategy, resources accounted for, and perspective. In a review of 34 studies of MDA costs, financial unit costs (excluding medicine) ranged from $0.01 to $8.50 (2015 USD).[16] Typically, financial costs for STH SBD have been estimated at less than $0.50 per treatment administered (USD between 1993 and 2007).[9] Our SBD routine financial costs align with these estimates; however, our total economic costs are generally higher, due to the inclusion of planning costs, opportunity costs for teachers and other government staff and supportive supervision and data collection activities. Few STH cMDA costs are available in the literature. The Tuangamize Minyoo Kenya Imarisha Afya (TUMIKIA) study in Kenya estimated total programme costs of biannual cMDA at $0.76 per treatment administered and routine programme costs at $0.50 (2016 USD, economic costs).[10] These routine cost estimates are similar to DeWorm3 routine costs in India ($0.42), though are lower than routine costs in Malawi ($0.78) and Benin ($1.13, all country results in 2019 USD, economic costs). Our cMDA unit costs are comparable to other studies evaluating economic costs of cMDA for NTDs, such as trachoma costs (estimated at $1.53, 2010 USD, excluding costs of antibiotics) and lymphatic filariasis costs (ranging from $0.40 to $5.87, USD between 2000 and 2009, including drug costs).[12–14]

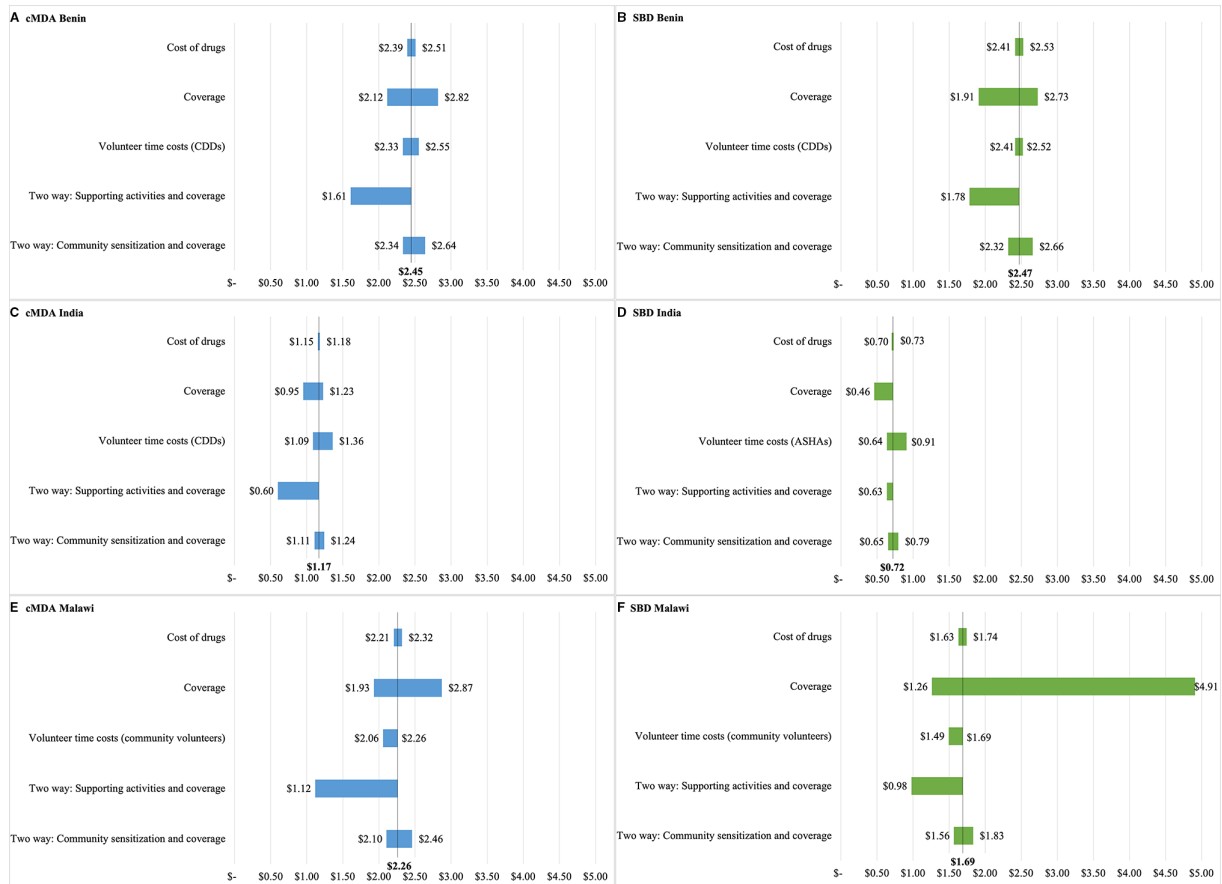

**Figure 2** One-way and two-way sensitivity analyses of unit costs (2019 USD ($)). (A) community-wide mass drug administration (cMDA) costs in Benin; (B) school-based deworming (SBD) costs in Benin; (C) cMDA costs in India; (D) SBD costs in India; (E) cMDA costs in Malawi; (F) SBD costs in Malawi. Details on how each parameter was varied can be found in online supplemental appendix 4.

This study disaggregates routine programme costs from supportive costs that are used to increase coverage (additional sensitisation, NGO supervision and electronic data collection). Average routine costs of cMDA were lower than SBD costs across countries. This is largely driven by salary costs for teachers and school directors who generally spend 1–3 days each year involved in SBD. Similar findings were observed in Niger, where deworming was delivered via SBD to children and via community-based treatment to children and at-risk adults (at fixed locations or their homes); unit costs of SBD were higher at $0.76 compared with $0.46 for community treatment (2005 USD). Differences in costs in Niger were attributed to CDDs treating more individuals than teachers.[15] Our results demonstrated that wages, per-diems and opportunity costs (eg, time costs) for staff represented the largest share of total costs, a finding that was consistent across all sites and both implementation strategies. Similarly, the TUMIKIA trial found 67.5% of cMDA costs for STH in Kenya were financial and opportunity costs for personnel.[10] These findings highlight the importance of fully accounting for costs associated with the delivery workforce, including teachers involved in SBD and volunteer drug distributors in cMDA.

As it is not possible to disentangle the precise impact of supportive activities on coverage, sensitivity analyses were conducted to explore the potential impact of reducing supportive activities on unit costs. If supportive activities were removed and coverage reduced as a result, unit costs were estimated to drop between 10% and 50%. Although opportunities for electronic data collection during MDA are increasing (eg, ESPEN Collect), not all programmes may choose to proceed with more resource-intensive mobile data collection.[27] However, evidence suggests high coverage of cMDA may be necessary to interrupt transmission, and, thus, the total costs presented in this study may be representative of costs incurred by elimination programmes.

Given the experimental nature of cMDA and the DeWorm3 platform on which it was implemented, cMDA costs may vary if launched within routine health systems. Depending on existing capacity within countries, governments could see a reduction in costs due to cost-sharing between other community-based or NTD programmes. Additionally, studies suggest that MDA costs are subject to economies of scale; according to one model, a 10-fold increase in individuals treated could reduce costs by approximately 70% in DeWorm3 countries.[11 16] Costs of

cMDA collected over the first 2 years of implementation in DeWorm3 may have been high due to start-up costs, and, therefore, costs could reduce over time with experience, as observed in Haiti's integrated STH and lymphatic filariasis MDA programme, which saw a decrease in cost per person treated from $2.23 during the first year of implementation in 2000 to $0.64 per person between 2008 and 2009 (USD).[14] Future modelled analyses of DeWorm3 costing data will explore costs of scaling cMDA programmes, altering frequencies and sampling strategies for conducting additional programme activities (eg, censuses, prevalence surveys and coverage surveys) and examining implications on drug costs if cMDA for STH was to be scaled up widely.

When examining average unit costs of cMDA and SBD across sites, we observed lowest costs in India, followed by Malawi and Benin, respectively. However, this pattern was not consistent when examining costs per round, by subactivity or by routine versus opportunity cost. For example, unit costs of cMDA were highest in Malawi during rounds 1 and 2. Our results suggest unit costs of planning, training and community sensitisation may be more similar across MDA treatment strategies and countries, while resources such as staffing and supervision for programme management and drug delivery may be more setting specific. We briefly highlight several reasons for variation in unit costs across sites and a more extensive description of drivers of variation is found in online supplemental appendix 6. Sites varied in the degree of NGO and government involvement. In Benin, the DeWorm3 team and the Ministry of Health worked closely together to implement cMDA and SBD. In Malawi, the DeWorm3 team led the implementation of both cMDA and SBD with supervisory support from the Government of Malawi. This close collaboration on implementation in Benin and Malawi incurred more allowances and opportunity costs for both 'supportive' NGO staff and 'routine' government staff. In India, there was a greater separation of responsibilities for cMDA and SBD, with the DeWorm3 team implementing cMDA and the Government of India implementing SBD. Given SBD was solely led by the Government of India, 'supportive' costs were substantially lower. A driver of heterogeneity in SBD costs was variation in school staff involvement across sites. Opportunity costs for school staff were higher in India and Benin given a larger number of school staff such as teachers, Anganwadi Workers and school directors involved, and higher salaries for school staff. Finally, the different number of treatments administered, due to population sizes, population age compositions and coverage rates, affected unit costs. For example, total costs of SBD were similar in Benin and Malawi, however, more school-aged children were treated in Malawi resulting in lower unit costs. Previous studies have similarly reported differences in unit costs across countries and wide intracountry variation. The TUMIKIA study reported average unit costs of biannual cMDA in Kenya varied from $0.49 to $1.85 across clusters (2016 USD).[10] Additionally, during nationwide scale-up of SBD in Uganda, costs varied $0.41—$0.91 across districts (2005 USD), given differences in number of children treated, community sensitisation costs and district-level supervision.[11] The intercountry and intracountry variations highlight the many ways STH treatment strategies can be implemented, and how community-based health campaigns may need to be adjusted to adapt to specific population needs. We encourage future STH MDA costing studies to report details of implementation costs and to explore drivers of variation in costs and coverage within and across countries.

In addition to unit costs, other metrics should be considered to determine the relative value of cMDA and SBD.[28] Cost-effectiveness analyses are important to compare costs to health benefits gained. If more children are treated through cMDA than SBD, and/or overall STH prevalence is reduced, costs per infection-year averted may be lower for cMDA compared with SBD. If cMDA interrupts STH transmission, the long-term reduction in STH burden as a result of cMDA could be substantial. After DeWorm3 trial results are unblinded, further analyses will determine the incremental cost-effectiveness of cMDA compared with SBD under multiple time horizons to account for the long-term benefits of elimination.

There are several limitations to this analysis. As described above, there were different degrees of DeWorm3 involvement in SBD across sites; data sources and some driving assumptions, thus necessarily varied. Although DeWorm3 trial costs were excluded from this costing analysis, we anticipate that programme management, planning and supervision costs may be higher than what would be observed routinely. Other assumptions are described in detail in online supplemental appendix 2.

## CONCLUSION

This study provides evidence from a large microcosting study, over 12 rounds of cMDA and 8 rounds of SBD in Benin, India and Malawi DeWorm3 sites. To our knowledge, this is the first study to directly compare costs of SBD to cMDA for STH programmes.[9] On average, cost per treatment administered through cMDA was more expensive than SBD in India and Malawi, and comparable in Benin. The largest difference in subactivity costs was related to drug delivery, where cMDA financial costs for routine resources (eg, CDD allowances) and supportive activities (eg, additional supervision) were notably higher than for SBD across all three countries. Although financial costs were higher for cMDA, opportunity costs for government-funded staff and volunteers were higher for SBD, mainly driven by teacher time. Overall, wages and per-diems represented the largest share of costs across countries and treatment strategies. Programme planners should consider what changes in

staffing and other resources are needed to implement cMDA at scale, knowing that costs may vary given cross-country differences and economies of scale. Future budget-impact and cost-effective analyses will generate additional evidence on the value for money and afford-ability of cMDA compared with SBD.

**Author affiliations**
¹Department of Global Health, University of Washington, Seattle, Washington, USA
²Institut de Recherche Clinique du Bénin, Abomey-Calavi, Bénin
³The Wellcome Trust Research Laboratory, Division of Gastrointestinal Sciences, Christian Medical College, Vellore, Tamil Nadu, India
⁴Blantyre Institute for Community Outreach, Blantyre, Malawi
⁵Department of Epidemiology and Public Health, Swiss Tropical and Public Health Institute, Basel, Switzerland
⁶Faculty of Medicine, University of Basel, Basel, Switzerland
⁷Université de Paris, MERIT, IRD, Paris, France
⁸Centre de Recherche pour la lutte contre les Maladies Infectieuses Tropicales (CReMIT/TIDRC), Université d'Abomey-Calavi, Abomey-Calavi, Bénin
⁹Clinical Research Department, London School of Hygiene & Tropical Medicine, London, UK
¹⁰Faculty of Infectious and Tropical Diseases, London School of Hygiene & Tropical Medicine, London, UK

**Acknowledgements** The authors wish to thank all of the study participants, communities and community leaders, national NTD program staff, and local, regional and national partners (Ministry of Health and Family Welfare, Delhi and the Directorate of Public Health, Chennai in India; Programme National de lutte contre les Maladies Transmissibles du Ministère de la Santé du Bénin; and the Ministry of Health, Republic of Malawi) who have participated in or supported the implementation of the DeWorm3 study. We also would like to acknowledge the work of all members of the DeWorm3 study teams and affiliated institutions. We give special thanks to Dr. Kumudha Aruldas, Monrenike Bada, Miranda Benson, Chikondi Chikotichalera, Dr. Sobana Devavaram, Sean Galagan, Katherine Halliday, Noel Joyce Mary Hillari, Hugo Legge, Providence Nindi, Chinnaduraipandi Paulsamy, Emily Pearman, Rajeshkumar Rajendiran, Sarah Smith and Elodie Yard for assisting with data collection, cleaning and interpretation.

**Contributors** KG, MS, KK, AJFL, MI, RLB, RP, SSRA, JW and ARM conceptualised the project and methodology; JW secured the funding. EA, SPK, JS, SPG, WL and PH collected the data. EA, SPK, JS, SPG, WL, KG, MS and CM curated the data. KG, MS and CM analysed the data. EA and MI were involved in the validation and interpretation of results from Benin; SPK and SPG were involved in validation and interpretation of results from India; JS, WL and KK were involved in validation and interpretation of results from Malawi. ARM and CM drafted the manuscript and all authors revised and approved the manuscript for content. ARM is the guarantor.

**Funding** This work was funded by the Bill & Melinda Gates Foundation (INV-022149). The funders had no role in study design, data collection, data analysis, data interpretation, or writing of the article. We received no compensation from pharmaceutical companies or other agencies to write this article.

**Competing interests** None declared.

**Patient and public involvement** Patients and/or the public were not involved in the design, or conduct, or reporting, or dissemination plans of this research.

**Patient consent for publication** Not applicable.

**Ethics approval** This study was affiliated with a trial that was reviewed and approved by the Institut de Recherche Clinique au Bénin (IRCB) through the National Ethics Committee for Health Research (002-2017/CNERS-MS) from the Ministry of Health in Benin, The London School of Hygiene and Tropical Medicine (12013), The College of Medicine Research Ethics Committee (P.04/17/2161) in Malawi, and the Institutional Review Board at Christian Medical College, Vellore (10392). The DeWorm3 Project was also approved by the University of Washington (STUDY00000180).

**Provenance and peer review** Not commissioned; externally peer reviewed.

**Data availability statement** Data are available upon reasonable request. Data collected from this study, including resources used, costs incurred, and the number of albendazole treatments administered, and a corresponding data dictionary

will be made available upon reasonable request to the corresponding author. For inquiries, contact Dr. Arianna R. Means at aerubin@uw.edu.

**ORCID iDs**
Chloe Morozoff http://orcid.org/0000-0002-3254-5553
James Simwanza http://orcid.org/0000-0003-0505-9952
Katya Galactionova http://orcid.org/0000-0002-5743-7647
Arianna Rubin Means http://orcid.org/0000-0002-4087-7080

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
