## [Reviewer comments · BMJ Open]

ARTICLE DETAILS

TITLE (PROVISIONAL)	Costs of community-wide mass drug administration and school-based deworming for soil-transmitted helminths: evidence from a randomized-controlled trial in Benin, India, and Malawi
AUTHORS	Morozoff, Chloe; Avokpaho, Euripide; PUTHUPALAYAM KALIAPPAN, SARAVANAKUMAR; Simwanza, James; Gideon, Samuel; Lungu, Wongani; Houngbegnon, Parfait; Galactionova, Katya; Sahu, Maitreyi; Kalua, Khumbo; Luty, Adrian J. F.; Ibikounle, Moudachirou; Bailey, Robin; Pullan, Rachel; Ajjampur, Sitara; Walson, Judd; Means, Arianna

VERSION 1 – REVIEW

REVIEWER	Richardson, Alice Australian National University, NCEPH
REVIEW RETURNED	16-Dec-2021

GENERAL COMMENTS	In this paper the authors report on the costs aspect of a cluster randomised trial across the three countries Benin, India and Malawi. The trial aims to compare SBD and cMDA as administration methods for deworming medication. Twenty sites were randomised to each arm of the study (SBD and cMDA). This paper would benefit from a statement justifying the sample size, so that readers do not need to dive into the other papers associated with this study in order to find this justification. Tables 2 and 3, and correspondingly Tables 8 and 9, contain dashes for a handful of cells. The authors should explain what the dash means – no population or no sample, for example. The costs derived from the analysis are given as point estimates, with no indication of standard errors to go with them. I wonder if this is a feature of economic analyses, that standard errors are not included, but I would like to suggest that the authors consider providing some measure of the variability expected in these estimates. Then readers would also be in a better position to tell whether the \$1.17 is significantly different from the \$2.26, for instance, or the \$2.26 from the \$1.69. Some costs in Table 5 have interval estimates included but this is not consistent. The authors could make better use of the range of values in table 5 to provide a range of values in their final estimates. Finally, I think the Figures in Appendix 5 are not yet publication worthy. In Figure 1, the pie charts are hard to compare, and the large slice dominates. The authors should consider replacing them with stacked bar charts. In Figure 2, I'm not sure that the mosaic bars offer better
--

	interpretation that stacked bar charts again. The authors should consider alternative presentation of the bars. I am also very concerned that the MDA data for Malawi has failed to take account of other costs. The authors should comment on this. In regards to Figure 3, I am not convinced that box and whisker plots should be used to display fewer than five data points. Here we have either four or two, and the boxplots conceal this. The authors should consider replacing the box and whisker plots with the actual data in a scatter plot or similar. The English expression in the paper is satisfactory and I found just one typo. Page 4 line 17: routine not routing. In the light of the above, I am recommending that the paper undergo major revision according to the points raised above, and then be resubmitted for reconsideration by BMJ Open.
--	--

REVIEWER	Masaku, Janet Kenya Medical Research Institute, Eastern and Southern Africa Centre for International Parasite Control
REVIEW RETURNED	09-Mar-2022

GENERAL COMMENTS	This study topic is of importance in the NTDs programme area. Nevertheless, a lot of assumptions have been left for the reader to do on how the study was done. There is need to improve the content of the manuscript.  1. All the full stops are appearing before the references in the all document, is this a new way of referencing? should be after the references. 2. Line spacing is quite small in the all document 3. The introduction quite brief, more information on Deworm 3 is needed, why the specific countries? What others have done in MDA costing, also indicate the current study will be affiliated to the main Deworm 3 clinical trial. 4. How many rounds of MDA were done? For how long? 5. Where the MDAs cover all the whole countries mentioned or was is part or some region? 6. Methods 7. How many schools and communities were involved in Deworm 3 in the three counties? 8. When was the Deworm 3 project started? 9. It would be good to give an insight on how the schools and communities were selected by the Deworm 3? Which method was used to select them
---

	during the costing? 10. The costing was done for 3 years, it would be good to justify why three years and not the entire period for the Deworm 3 project though not mentioned 11. How many rounds of MDA were done for the costing? 12. Results -Line 32 of the results section, the costing seems to have been done in the months mentioned why was not this information introduced in the methodology? In another section in the methodology, it was done after 3 years? There is inconsistency. 13. Discussion- 14. Line 37 and 38 references 10 and 12 are on superscript. Also, the variations in the TUMIKIA study in Kenya and Uganda, could there be an explanation? 15. The discussion needs to be a bit elaborate. The reader is left not sure why the costs are higher in one site and not the other. 16. It worth to note this area of NTDs programme costing has not been researched much. Nevertheless, there is need to give the reader more information of what others have found especially in line with the study findings and study topic. 17. Conclusion- Line 5 and 6, that information could have been given in the introduction 18. Line 14; Overall, wages and per-diems represented the largest share of costs across countries and treatment strategies. I have not seen these findings being discussed in the discussion section
--	---

VERSION 1 – AUTHOR RESPONSE

Reviewer #1:

In this paper the authors report on the costs aspect of a cluster randomised trial across the three countries Benin, India and Malawi. The trial aims to compare SBD and cMDA as administration methods for deworming medication.

Twenty sites were randomised to each arm of the study (SBD and cMDA). This paper would benefit from a statement justifying the sample size, so that readers do not need to dive into the other papers associated with this study in order to find this justification.

The number of persons treated by SBD and cMDA is reported in our article and used for the assessment of unit costs. We included citations for trial protocols, if readers are interested in further understanding the DeWorm3 trial and reading more details about the approach used for sample size calculation. We feel that sample size of the underlying randomized-controlled trial is not directly relevant to the costing analysis, as the goal of the paper is to describe costs of cMDA and SBD as implementation strategies.

However, to provide more information to the reader about the size and scale of the study, we have added details about cluster size to the methods, and the total number of persons enumerated in the DeWorm3 study clusters at baseline to Appendix Table 5.

In each site, twenty control clusters (minimum population size of 1,650 persons per cluster) were randomized to SBD (either annually or bi-annually, per the country's standard of care) and twenty intervention clusters were randomized to biannual cMDA.

Tables 2 and 3, and correspondingly Tables 8 and 9, contain dashes for a handful of cells. The authors should explain what the dash means – no population or no sample, for example.

Dashes represent situations where no data were collected, often due to no costs being incurred or no activities being implemented. We have added footnotes to tables for clarity.

The costs derived from the analysis are given as point estimates, with no indication of standard errors to go with them. I wonder if this is a feature of economic analyses, that standard errors are not included, but I would like to suggest that the authors consider providing some measure of the variability expected in these estimates. Then readers would also be in a better position to tell whether the \$1.17 is significantly different from the \$2.26, for instance, or the \$2.26 from the \$1.69. Some costs in Table 5 have interval estimates included but this is not consistent. The authors could make better use of the range of values in table 5 to provide a range of values in their final estimates.

We appreciate the reviewer's suggestion to improve the representation of variability in mean unit costs. In our analysis, we observed costs per site (Come Commune of Benin, Tamil Nadu State of India, and Mangochi District of Malawi), strategy (cMDA and SBD), and round of treatment. The range of persons treated, total costs, and unit costs observed are presented in Table 1. We further explored variability in unit costs through sensitivity analyses in Figure 2.

In Tables 2 and 3, we reported mean unit costs across rounds, per site and strategy. Given the large amount of data presented in Tables 2 and 3 (78 data points per table), we feel it would be challenging for readers to interpret costs if the range of costs across rounds were added to each cell. However, to better demonstrate how the breakdown of sub-activity costs (e.g., unit costs for program management, planning, community sensitization, training, MDA, mop-up) varied across rounds, we have generated a plot, which has been added as Appendix Figure 2. This plot describes the minimum, maximum, and mean unit cost per sub-activity, broken down by cMDA vs SBD, and by country. We have also added some additional text to describe this figure to the results section:

When examining how unit costs per sub-activity varied across rounds, actual MDA delivery costs were the most variable across sites and rounds, followed by program management costs (Appendix Figure 2). After planning costs, which were annualized across rounds, community sensitization showed the least amount of variability in unit costs across countries, rounds, and treatment strategies.

Additionally, we have added additional text to the discussion:

Our results suggest unit costs of planning, training and community sensitization may be more similar across MDA treatment strategies and countries, while resources such as staffing and supervision for program management and drug delivery may be more setting specific.

The scope of this study was to present costs of cMDA and SBD as observed in the DeWorm3 project sites. Future analyses may consider variability in programmatic inputs to further generalize and model costs outside of the DeWorm3 sites.

Finally, I think the Figures in Appendix 5 are not yet publication worthy. In Figure 1, the pie charts are hard to compare, and the large slice dominates. The authors should consider replacing them with stacked bar charts. In Figure 2, I'm not sure that the mosaic bars offer better interpretation than stacked bar charts again. The authors should consider alternative presentation of the bars. I am also very concerned that the MDA data for Malawi has failed to take account of other costs. The authors should comment on this.

Thank you for your suggestion. We have replaced Figures 1 and 2 with a stacked bar chart (now Appendix Figure 1).

"Other" costs were in fact considered in Malawi. These costs were minor and represented less than 0.5% of total costs, therefore are rounded down and presented as 0% costs. We have added a footnote to Appendix Figure 1 for clarity.

For simplicity, labels were rounded to the nearest whole number. In situations where 0% is listed, category costs contributed less than 0.5% of total costs.

In regards to Figure 3, I am not convinced that box and whisker plots should be used to display fewer than five data points. Here we have either four or two, and the boxplots conceal this. The authors should consider replacing the box and whisker plots with the actual data in a scatter plot or similar.

Thank you for this suggestion. We have replaced Appendix Figure 3 with a new plot (now Appendix Figure 2), describing the minimum, maximum, and mean costs across rounds and countries.

The English expression in the paper is satisfactory and I found just one typo. Page 4 line 17: routine not routing.

Thank you for identifying this typo. We have corrected it.

Reviewer #2:

This study topic is of importance in the NTDs programme area. Nevertheless, a lot of assumptions have been left for the reader to do on how the study was done. There is need to improve the content of the manuscript. See file attached for comments.

Thank you for your feedback. We have updated the manuscript per your suggestions and provided detailed responses below.

This costing analysis was embedded within the DeWorm3 study, a cluster randomized-controlled trial to evaluate the effect of community-wide mass drug administration (cMDA) on soil-transmitted helminth (STH) prevalence, compared to school-based deworming (SBD). The DeWorm3 study also included several implementation science activities to assess feasibility of implementing cMDA across countries. Given the focus of this paper is on the costs of two implementation strategies for STH, rather than on the trial component of the DeWorm3 study, we did not include extensive information about the parent cluster RCT in the main text of this paper. Main findings from the parent trial are pending and will be published separately.

However, to increase the readability of the paper, we have added additional details about the parent study to the main text and the appendix. For additional information on the DeWorm3 study design and contexts, readers are also referred to the published cluster RCT and implementation science protocols.

All the full stops are appearing before the references in the all document, is this a new way of referencing? should be after the references.

Per BMJ Open's formatting guidelines, citations should appear after punctuation, not before: "Reference numbers in the text should be inserted immediately after punctuation (with no word spacing)—for example,[6] not [6]."

Line spacing is quite small in the all document

We have updated the line spacing to 1.5 for your review. If accepted for publication, the manuscript will be reformatted by BMJ Open's editorial team to align with journal formatting.

The introduction quite brief, more information on Deworm 3 is needed, why the specific countries? What others have done in MDA costing, also indicate the current study will be affiliated to the main Deworm 3 clinical trial.

This feedback from the reviewer is well received. We have expanded the methods section text, under the subtitle "Overview of DeWorm3" to provide more information about the parent study. For example, we have added:

The DeWorm3 project was implemented in Come Commune of Benin, Tamil Nadu State of India, and Mangochi District of Malawi. These sites were selected because they had previously implemented lymphatic filariasis programs over five or more rounds with albendazole co-administered with ivermectin or diethylcarbamazine.[8]

DeWorm3 trial sites were selected based on historical MDA programs for lymphatic filariasis. We have attempted to further clarify that the current study is affiliated with the main DeWorm3 trial by updating paragraph 4 of the introduction. Also of note, once outcome assessments are available for DeWorm3, an additional cost-effectiveness analysis will be conducted and published separately to describe the incremental cost-effectiveness of implementing community-wide MDA compared to the standard of care.

This study systematically identified, measured, and compared resources for implementation of twelve rounds of cMDA and eight rounds of SBD across the DeWorm3 sites, during implementation of the DeWorm3 trial.

At the reviewer's suggestion, we have expanded upon information provided about SBD costs in paragraph 2 with additional information about costing initiatives for cMDA in paragraph 3 of the introduction:

Although studies have evaluated the costs of mass drug administration for neglected tropical diseases, costs vary based on country implementation strategy (e.g., use of volunteers or salaried staff), disease control program, age of program, size of population treated, and costing methods used.[10-18] To our knowledge, there are no studies that directly compare costs of cMDA and SBD for STH control, within the same setting and methodological framework.

How many rounds of MDA were done? For how long?

cMDA was conducted biannually across all three countries. SBD was implemented differently across countries; in India SBD was conducted biannually while in Benin and Malawi SBD was conducted annually. The DeWorm3 project was conducted for three years, however only two years of costing data were collected, resulting in costs for 12 total rounds of cMDA and 8 rounds of SBD (see Table 1). We have added additional clarification to the methods:

Across the three sites, we conducted two years of intensive micro-costing, resulting in data from 12 rounds of cMDA and 8 rounds of SBD. Costing the first year of cMDA implementation allowed us to capture costs related to start-up, while the second year provided a more accurate portrayal of costs related to routine implementation.

The number of days needed for drug delivery varied across countries and rounds. Average, minimum, and maximum number of days of cMDA and SBD drug delivery per site are described in Appendix Table 5.

Where the MDAs cover all the whole countries mentioned or was is part or some region?

The DeWorm3 project was implemented only in select areas, described in Appendix Table 1. We have added a sentence to the methods for readers to ensure clarity:

The DeWorm3 project was implemented in Come Commune of Benin, Tamil Nadu State of India, and Mangochi District of Malawi... In each site, twenty control clusters (minimum population size of 1,650 persons) were randomized to ...

How many schools and communities were involved in Deworm 3 in the three counties?

The number of schools involved in each site are detailed in Appendix Table 5. Each cluster included a minimum population size of 1,650 persons, which consisted of one or more villages, settlements, or zones within urban areas. We have updated the Appendix Table 5 to also include the number of villages involved, and have added updated the manuscript text as follows:

Number of schools, villages, and other site-level contextual attributes are described in Appendix Tables 1, 2 and 5.

When was the Deworm 3 project started?

The DeWorm3 project was launched in 2015, the first census was conducted in 2017, and the first round of MDA was in 2018. We have added the dates of MDA to the methods.

During the DeWorm3 project, cMDA and SBD were implemented for three years, from 2018-2020.

It would be good to give an insight on how the schools and communities were selected by the Deworm 3? Which method was used to select them during the costing?

The DeWorm3 study was conducted in three sites (Come Commune of Benin, Tamil Nadu State of India, and Mangochi District of Malawi). Sites were selected because they had previously implemented lymphatic filariasis programs over five or more rounds using albendazole with ivermectin or diethylcarbamazine. In each site, twenty clusters were randomized to cMDA, and twenty clusters were randomized to SBD. Each cluster contained a minimum population size of 1,650 persons, and all communities and schools within the clusters participated in the trail.

Costing of SBD and cMDA was done across the entire DeWorm3 study site. Therefore, all schools and communities located within the study clusters were included in the costing study.

The costing was done for 3 years, it would be good to justify why three years and not the entire period for the Deworm 3 project though not mentioned

We have added this information to the methods:

Across the three sites, we conducted two years of intensive micro-costing, resulting in data from 12 rounds of cMDA and 8 rounds of SBD. Costing the first year of cMDA implementation allowed us to capture costs related to start-up, while the second year provided a more accurate portrayal of costs related to routine implementation.

How many rounds of MDA were done for the costing?

In total, we costed 12 total rounds of cMDA (4 rounds per country) and 8 rounds of SBD (2 rounds in Benin, 4 rounds in India, and 2 rounds in Malawi. Costs for each country-round are included Table 1. We have added additional clarification to the methods section:

Across the three sites, we conducted two years of intensive micro-costing, resulting in data from 12 rounds of cMDA and 8 rounds of SBD.

Line 32 of the results section, the costing seems to have been done in the months mentioned why was not this information introduced in the methodology? In another section in the methodology, it was done after 3 years? There is inconsistency.

In total, MDA was completed for 3 years during the DeWorm3 project, however costing was only conducted for the first 2 years of MDA. The dates in the results section are describing specific dates of MDA that were costed.

We have clarified by adding a sentence to the *Overview of DeWorm3 project* section stating:

During the DeWorm3 project, cMDA and SBD were implemented for three years, from 2018-2020. We have also added some additional context to explain why costing was conducted for two years (as opposed to the full three years of the trial), under the *Costing study design* section:

We conducted activity-based micro-costing from the service provider perspective (Ministry of Health and/or Education) during the first two years of DeWorm3 implementation in order to explore heterogeneity in costs across rounds of implementation. Across the three sites, we conducted two years of intensive micro-costing, resulting in data from 12 rounds of cMDA and 8 rounds of SBD. Costing the first year of cMDA implementation allowed us to capture costs related to start-up, while the second year provided a more accurate portrayal of costs related to routine implementation.

Line 37 and 38 references 10 and 12 are on superscript. Also, the variations in the TUMIKIA study in Kenya and Uganda, could there be an explanation?

Thank you, we have fixed the reference format. For the TUMIKIA study in Kenya, cost variation across clusters is not explained. The Uganda study provides some suggested reasons for variation, including differences in number of children treated, differences in community sensitization costs, and difference in number of days district officials were involved in supervision across districts. This is similar to our analysis, where we highlight the ways that implementation may vary naturally to address population needs, and the resulting effect on variability in costs. In paragraph 6 of the discussion, we describe drivers of variability in costs across countries involved in the DeWorm3 project. We have updated this paragraph to include the reasons for variation in Uganda.

Previous studies have similarly reported differences in unit costs across countries, and wide intra-country variation. The TUMIKIA study reported average unit costs of biannual cMDA in Kenya varied from \$0.49—\$1.85 across clusters.[10] Additionally, during nationwide scale-up of SBD in Uganda, costs varied \$0.41—\$0.91 across districts, given differences in number of children treated, community sensitization costs, and district-level supervision.[11] The inter- and intra-country variations highlight the many ways STH treatment strategies can be implemented, and how community-based health campaigns may need to be adjusted to adapt to specific population needs.

The discussion needs to be a bit elaborate. The reader is left not sure why the costs are higher in one site and not the other.

We have elaborated further on reasons for differences across sites. Please also note that an extensive list of variations across sites- and thus potential cost drivers-

are detailed in Appendix 6. Given word count limits, we are not able to go into all these extensive details within the main text of the paper.

When examining average unit costs of cMDA and SBD across sites, we observed lowest costs in India, followed by Malawi, and Benin respectively. However, this pattern was not consistent when examining costs per round, by sub-activity, or by routine vs. opportunity cost. For example, unit costs of cMDA were highest in Malawi during rounds 1 and 2. Our results suggest unit costs of planning, training and community sensitization may be more similar across MDA treatment strategies and countries, while resources such as staffing and supervision for program management and drug delivery may be more setting specific. We briefly highlight several reasons for variation in unit costs across sites and a more extensive description of drivers of variation can be found in Appendix 6. Sites varied in the degree of NGO and government involvement. In Benin, the DeWorm3 team and the Ministry of Health worked closely together to implement cMDA and SBD. In Malawi, the DeWorm3 team led the implementation of both cMDA and SBD with supervisory support from the Government of Malawi. This close collaboration on implementation in Benin and Malawi incurred more allowances and opportunity costs for both “supporting” NGO staff and “routine” government staff. In India, there was a greater separation of responsibilities for cMDA and SBD, with the DeWorm3 team implementing cMDA and the Government of India implementing SBD. Given SBD was solely led by the Government of India, “supporting” costs were substantially lower. A driver of heterogeneity in SBD costs was variation in school staff involvement across sites. Opportunity costs for school staff were higher in India and Benin given a larger number of school staff such as teachers, Anganwadi Workers, and school directors involved, and higher salaries for school staff. Lastly, the different number of treatments administered, due to population sizes, population age-compositions, and coverage rates, affected unit costs. For example, total costs of SBD were similar in Benin and Malawi, however, more school-aged children were treated in Malawi resulting in lower unit costs.

It worth to note this area of NTDs programme costing has not been researched much. Nevertheless, there is need to give the reader more information of what others have found especially in line with the study findings and study topic.

Thank you for this feedback. Our discussion section includes comparative information on other MDA costs for NTDs. We have expanded this content throughout the entire discussion, so it is given greater focus. We have also added a call for future STH costing studies:

We encourage future STH MDA costing studies to report details of implementation costs and to explore drivers of variation in costs and coverage within and across countries.

Line 5 and 6, that information could have been given in the introduction

Thank you for your suggestion, we have now included this information earlier in the manuscript. We felt this information fit most naturally in the methods section, under the section “costing study design”, where we describe details of the study:

Across the three sites, we conducted two years of intensive micro-costing, resulting in data from 12 rounds of cMDA and 8 rounds of SBD... We measured all resources required to deliver cMDA and SBD in DeWorm3 clusters, resulting in over 8,000 data points, and converted their value into a cost estimate (including borrowed and donated resources).

Line 14; Overall, wages and per-diems represented the largest share of costs across countries and treatment strategies. I have not seen these findings being discussed in the discussion section

Thank you for highlighting this point. You are correct that this information was mentioned in the results, but not the discussion. We have now added a sentence to the discussion to this point:

Our results demonstrated that wages, per-diems, and opportunity costs (ex. time costs) for staff represented the largest share of total costs, a finding that was consistent across all sites and both implementation strategies.

VERSION 2 – REVIEW

REVIEWER	Richardson, Alice Australian National University, NCEPH
REVIEW RETURNED	05-May-2022
GENERAL COMMENTS	In this revision the authors have responded to all the points raised in my referee's report. In particular, they have replaced Figure 1 in Appendix 5 which was a set of pie charts with a set of stacked bar charts, which are much more interpretable. They have also removed Figure 2 and replaced Figure 3 which was a set of boxplots with a set of "box and whisker" plots on a new unit cost scale for ready comparison. I would like to thank the authors for their consideration of how best to display the information in the old Figure 3, while also taking the opportunity to extend it to multiple countries. In the light of the above, I am recommending that the paper be accepted for publication by BMJ Open.